# Improving Primary Care After Stroke (IPCAS) trial: protocol of a randomised controlled trial to evaluate a novel model of care for stroke survivors living in the community

Ricky Mullis,[1] Maria Raisa Jessica (Ryc) Aquino,[1] Sarah Natalie Dawson,[2] Vicki Johnson,[3] Sue Jowett,[4] Elizabeth Kreit,[1] Jonathan Mant,[1] on behalf of the IPCAS investigator team

¹General Practice and Primary Care Research Centre, University of Cambridge, Cambridge, UK
²MRC Biostatistics Unit, University of Cambridge Institute of Public Health, Cambridge, UK
³Leicester Diabetes Centre, University Hospitals of Leicester NHS Trust, Leicester, UK
⁴Health Economics Unit, University of Birmingham, Birmingham, UK

**Correspondence to**
Dr Ricky Mullis;
r.mullis@medschl.cam.ac.uk

## ABSTRACT

**Introduction** Survival after stroke is improving, leading to increased demand on primary care and community services to meet the long-term care needs of people living with stroke. No formal primary care-based holistic model of care with clinical trial evidence exists to support stroke survivors living in the community, and stroke survivors report that many of their needs are not being met. We have developed a multifactorial primary care model to address these longer term needs. We aim to evaluate the clinical and cost-effectiveness of this new model of primary care for stroke survivors compared with standard care.

**Methods and analysis** Improving Primary Care After Stroke (IPCAS) is a two-arm cluster-randomised controlled trial with general practice as the unit of randomisation. People on the stroke registers of general practices will be invited to participate. One arm will receive the IPCAS model of care including a structured review using a checklist; a self-management programme; enhanced communication pathways between primary care and specialist services; and direct point of contact for patients. The other arm will receive usual care. We aim to recruit 920 people with stroke registered with 46 general practices. The primary endpoint is two subscales (emotion and handicap) of the Stroke Impact Scale (SIS) as coprimary outcomes at 12 months (adjusted for baseline). Secondary outcomes include: SIS Short Form, EuroQol EQ-5D-5L, ICEpop CAPability measure for Adults, Southampton Stroke Self-management Questionnaire, Health Literacy Questionnaire and medication use. Cost-effectiveness of the new model will be determined in a within-trial economic evaluation.

**Ethics and dissemination** Favourable ethical opinion was gained from Yorkshire and the Humber-Bradford Leeds NHS Research Ethics Committee. Approval to start was given by the Health Research Authority prior to recruitment of participants at any NHS site. Data will be presented at national and international conferences and published in peer-reviewed journals. Patient and public involvement helped develop the dissemination plan.

**Trial registration number** NCT03353519

### Strengths and limitations of this study

► This research is an evaluation of a systematically developed complex intervention.
► The trial is a randomised controlled design with broad inclusion criteria to maximise generalisability.
► Economic evaluation will determine the cost-effectiveness of the intervention.
► Due to the pragmatic nature of this trial only limited blinding of the research team to treatment allocation is possible.
► Exclusion of nursing home residents will restrict the relevance of the findings for this subgroup of stroke survivors.

## BACKGROUND AND RATIONALE

Survival after stroke is improving[1 2] leading to increased demand on primary care services to meet the long-term care needs of people with stroke living in the community. Surveys suggest these needs are not being adequately addressed and that many stroke survivors are dissatisfied with care after discharge from hospital.[3 4] Approximately a third of stroke survivors have moderate to severe levels of disability at 6 months.[5] In addition to the many physical consequences of stroke, commonly reported areas of concern include information needs, feelings of abandonment, problems with communication,[3] emotional, psychological and social problems, fatigue, and cognitive sequelae including poor memory and concentration.[6]

Little evidence exists as to how best to support long-term stroke survivors[7] especially beyond the first year after stroke,[8] and recent trials of greater specialist input after discharge from hospital have had mixed results.[7 9] No formal primary care-based holistic model of

care with clinical trial evidence exists to support stroke survivors living in the community, and stroke survivors report that many of their needs are not being met. Systematic reviews have demonstrated that self-management after stroke shows promise, but evidence on aspects such as mood and social tasks remains sparse, with wide CIs around effects on outcomes such as quality of life.[10–12]

Primary care could play an important role in the care of people with stroke, including secondary prevention and risk factor management, supporting access to community services, facilitating transfer back to specialist services, and education and provision of information about stroke. However, the feeling of 'abandonment' of people with stroke after hospital discharge suggests this role is not being fulfilled. Indeed, current recommendations,[13] such as for a structured review of needs beyond the first 6 weeks after discharge, are not being implemented.[14]

We have developed a novel multifactorial primary care model to address the longer term needs of stroke survivors living in the community. The components of the model have been assessed for feasibility of delivery within primary care across four general practices prior to starting the Improving Primary Care After Stroke (IPCAS) trial. This led to several minor procedural amendments aimed at improving implementation of the intervention.

## AIMS
The IPCAS trial aims to evaluate the clinical and cost-effectiveness of a new model of primary care for stroke survivors living in the community compared with standard care.

The primary endpoint for the trial will be two subscales (emotion and handicap) of the Stroke Impact Scale (SIS v3.0)[15] as coprimary outcomes at 12 months (adjusted for baseline).

### Trial design
Two-arm cluster-randomised controlled trial with general practice as the unit of randomisation.

Randomisation will be performed as random permuted block randomisation with a 1:1 allocation stratified by practice size.

### Setting and participants
General practices with a stroke register comprising a minimum of 100 patients and representing a range of urban/rural and different socioeconomic status from the East of England and the East Midlands will be recruited. We aim to recruit approximately 920 people with a confirmed history of stroke registered with 46 general practices. Given that primary care addresses the long-term needs of stroke survivors we did not restrict participants to any specific time interval after their stroke.

### Inclusion criteria
► On practice register with a history of stroke.

► Able to provide written informed consent (with or without the help of a carer).

► Age 18 years or older.

### Exclusion criteria
► Patients on the palliative care register.

► Living in a nursing home.

## METHODS
### Recruitment of stroke survivors
Prior to practice-level randomisation (see below) electronic searches of the clinical computer system will generate a list of people with a history of stroke who meet the inclusion criteria for the study.

Potentially eligible participants will be sent an invitation by their general practitioner (GP) to take part in the study. If practices had 110 or fewer such people, invitations were sent to all those eligible. For larger practices, a random sample of 110 eligible patients were sent invitations. The invitation pack contains an invitation cover letter, the Patient Information Sheet, consent form (online supplementary appendix 1), a questionnaire containing the coprimary outcomes and instructions to return the consent form and questionnaire to the researchers in a prepaid envelope (provided). If no response is received within 2 weeks from the initial mail-out, the practice will send a reminder. If no response is received after the reminder then no further attempts at contact will be made.

### Randomisation
Once all invitation letters and reminders have been sent out to patients within a practice, the practice will be randomised to intervention or control (ratio of 1:1). Randomisation will be performed centrally by the trial statistician using a stratified, random permuted block design. The stratification factor will be GP practice size, split into two levels: ≤10 500 and >10 500 patients, which reflects the median GP list size in the catchment area. The IPCAS trial flow chart can be seen in figure 1.

### Intervention
The new model of care incorporates a multifaceted package of service aimed at providing a structured review of stroke care needs, a self-management programme for survivors and their carers, optimised communication between patients and healthcare services, enhanced communication pathways between the different care services and increased awareness of and access to national and local community and charity-provided services. A logic model depicting the rationale for the IPCAS trial intervention can be seen in figure 2.

#### Structured review of patient needs
A structured review will be performed by a practice nurse or other appropriately trained member of the practice team. Consenting patients will be invited for review by the practice in the same way that they would normally be contacted (eg, post, telephone or short message

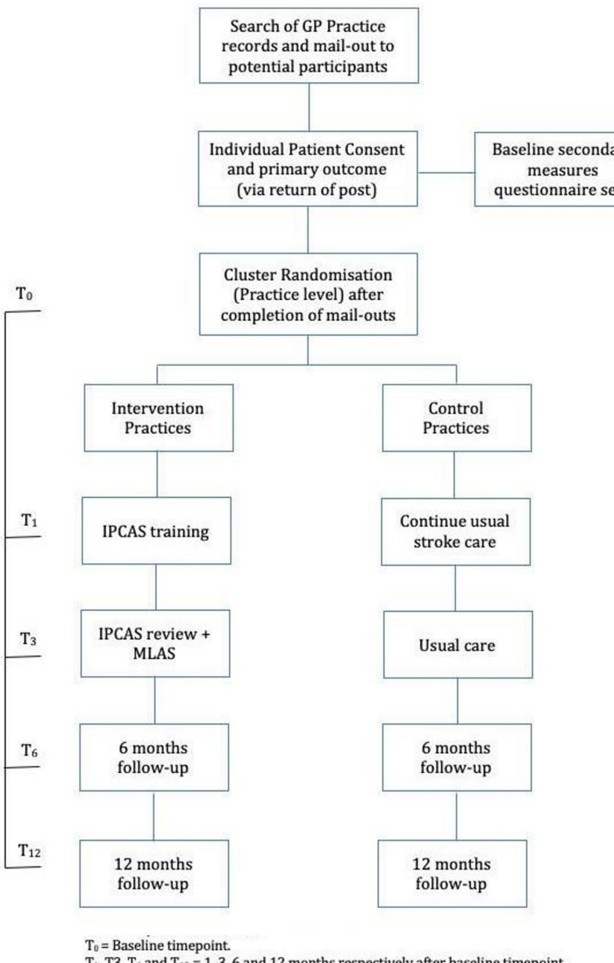

T₀ = Baseline timepoint.
T₁, T3, T₆ and T₁₂ = 1, 3, 6 and 12 months respectively after baseline timepoint.

**Figure 1** IPCAS trial flow chart. GP, general practice; IPCAS, Improving Primary Care After Stroke; MLAS, My Life After Stroke.

service). Where practicable, this review will be incorporated into the regular annual review recommended by current guidelines.[13] A 15-item checklist of common post-stroke needs[16] adapted from a checklist recommended by the World Stroke Organisation (WSO)[17] will be sent to the stroke survivor in advance, who will be asked to tick all needs which apply to them, and to bring this to the appointment. At the review, the patient will be asked which of the ticked items is their priority for immediate attention. Practice staff will discuss and address up to three key needs prioritised by the patient.

The review will last approximately 20–30 min and may include a routine physical check-up (eg, blood pressure, record of immunisation and medication review dependent on normal clinical practice at the GP surgery) followed by the discussion of poststroke care needs as identified by the stroke survivor. The outcome of the review will be an action plan agreed with the stroke survivor on how to address each of the key needs identified in the review.

The patient will be provided with an information leaflet introducing the self-management programme, with instructions on how to get further information and how to access the programme.

### Self-management programme (My Life After Stroke)

'My Life After Stroke' (MLAS) is a theory-driven self-management education programme with an explicit philosophical underpinning for stroke survivors and their carers (where appropriate) consisting of an initial individual preparatory session, four weekly group-based sessions and a final individual session. Individual appointments last approximately 30–45 min. Group sessions will include stroke survivors and their carers (where relevant) and last approximately 2.5 hours (including breaks).

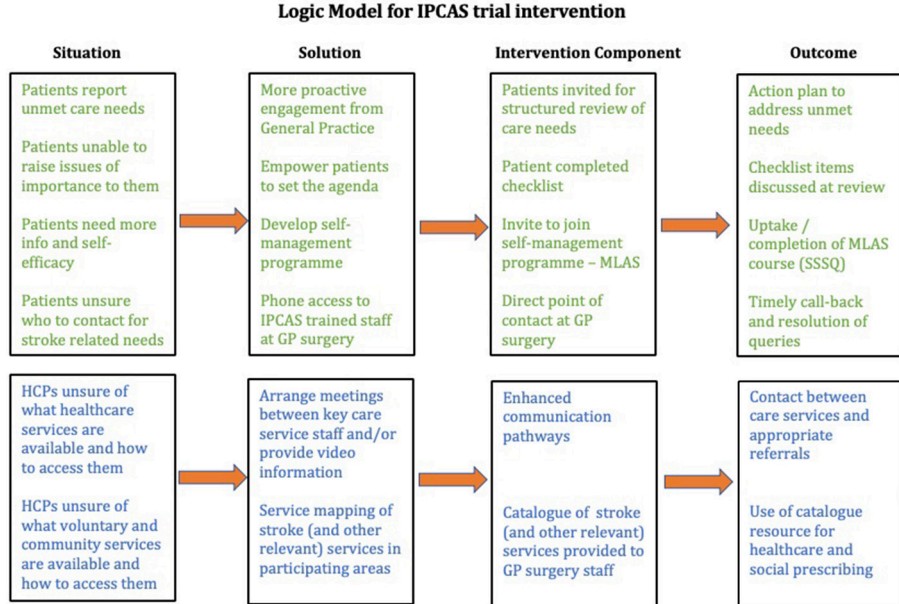

**Figure 2** Logic model for the IPCAS trial intervention. GP, general practitioner; HCP, healthcare practitioner; IPCAS, Improving Primary Care After Stroke; SSSQ, Southampton Stroke Self-management Questionnaire.

Group sessions cover a variety of topics including risk factors for stroke and prevention, psychological well-being, information, social needs, problem solving and goal setting. Participants will be given a handbook containing educational content and further information based on the session topics.

The programme will be run by two trained facilitators (healthcare professionals or people working in the voluntary sector) with an interest in or experience of stroke. All sessions will be held at a suitable, accessible, local community facility.

### Direct point of contact

A direct point of contact at the GP practice will be provided for stroke survivors and their carers. The staff member conducting the enhanced annual review will explain how to access the direct point of contact. Survivors or carers will be able to call the practice and indicate that they would like to talk to someone about a stroke-related problem. A single or several practice nurses or other appropriately trained healthcare members of the practice team will assume the role. If none of these people are available at the time of the call, a designated member of the care team will phone back. The aim of the direct point of contact will be to provide support and advice for stroke-specific issues, arranging follow-up appointments and signposting to further specialist or community services.

### Enhanced communication pathways

We will arrange a meeting between primary care staff from several practices and specialist staff (hospital and community) to facilitate primary/secondary care communication going forward. All practice staff involved with the care of stroke survivors will be encouraged to attend additional training/meetings organised by the specialist services, and given direct contact details for informal communication. Video recordings of local specialist(s) describing their service, the type of patients normally referred to the service and ways of contacting the service will be made available to all general practice staff.

### Service mapping

To support the information needs regarding local services for stroke-related problems, the care team will be provided with a catalogue of stroke (and other relevant) services in participating localities, including information on how to access them. This resource will be available in several electronic and hard copy formats to enable easy access by staff at the practice.

### Training for general practice staff

Training for practice staff involved in structured stroke reviews will include an overview of stroke and stroke-related long-term needs, followed by discussion of vignettes based on items from the stroke review checklist. Practice staff will suggest and discuss with the research team the most suitable course of action in each situation tailored to local context.

The list of key health and social services available in the local area will be provided, and practice staff will be familiarised with the service mapping resource that will be made available to them at the practice. The outcomes of the structured review will be recorded on a template in the patient records. We will discuss with the practice how best to embed the direct point of contact role within the current practice operations. To enable ease of attendance the training will be held in the practice and will last approximately 2 hours.

### Control arm

General practices randomised to the control arm of the trial will continue to deliver usual stroke care. Currently no standard package of long-term care for stroke survivors exists in primary care, and therefore we expect 'usual care' to vary between practices. We will capture information on the key elements of care provided by each participating practice to enable comparison between the two arms of the trial.

*Demographic data*: age, gender, ethnicity and postcode will be collected via postal questionnaire at the time of invitation to the study.

### Primary outcome

The primary endpoint for the trial will be two subscales (emotion and handicap) of the SIS v3.0[15] as coprimary outcomes at 12 months (adjusted for baseline) after randomisation of the practice.

### Secondary outcomes

To be collected at baseline, 6 and 12 months (*Collected at 12 months' follow-up only):
► SIS Short Form.[15]
► EuroQol EQ-5D-5L.[18]
► ICEpop CAPability measure for Adults (ICECAP-A).[19]
► Time since stroke.
► * Comorbidity, medication use (prescription and 'over the counter').
► * Southampton Stroke Self-management Questionnaire.[20]
► * Health Literacy Questionnaire.[21]

### Data collection

In this pragmatic, practice-level, cluster-randomised trial blinding to treatment allocation of the research team or clinical staff involved in delivering the intervention or control condition is not possible. The primary outcome will be captured by postal questionnaires sent to participants. Only in the event of missing data from the primary outcome will participants be contacted by the research team to either encourage them to return their questionnaire or to complete missing items via telephone. Questionnaire data entry onto an electronic spreadsheet will be outsourced to a third-party provider via secure data transfer for blinded data entry. The 'coded-allocation' spreadsheet will then be returned to the trial statistician, who will undertake all analyses independent of the rest of the research team.

| | STUDY PERIOD | | | |
|---|---|---|---|---|
| | Enrolment | Allocation | Post-allocation | |
| TIMEPOINT | -t₁ | 0 | 6 months | 12 months |
| **ENROLMENT:** | | | | |
| Informed consent | X | | | |
| Allocation | | X | | |
| **INTERVENTIONS:** | | | | |
| *[Intervention]* | | | ●———————————● | |
| *[Control]* | | | ●———————————● | |
| **ASSESSMENTS:** | | | | |
| *[Baseline variables]* | X | | | |
| *[Primary outcomes]* | | | X | X |
| *[Secondary outcomes]* | | | X | X |

**Figure 3** Improving Primary Care After Stroke (IPCAS) trial Standard Protocol Items: Recommendations for Interventional Trials (SPIRIT) flow chart showing scheduled enrolment, interventions and assessments of participants.

Baseline: The primary outcome data (emotion and handicap subscales of the SIS) will be collected via postal questionnaire at the time of invitation to the study prior to randomisation of the practice. Secondary outcome data (SIS Short Form, EuroQol EQ-5D-5L and ICECAP-A) will be collected by postal questionnaire after receipt of consent. Non-responders to the secondary outcome questionnaire will be followed up by telephone or the most appropriate method for a participant with aphasia.

Follow-up: At 6 and 12 months by postal questionnaire. Non-responders/incomplete responders will be followed up by telephone or the most appropriate method for a participant with aphasia.

A review of the GP notes of consenting participants will be conducted. Data extracted will include number and nature of primary care visits, secondary care inpatient and outpatient visits, investigations, medications and use of social services.

The IPCAS trial Standard Protocol Items: Recommendations for Interventional Trials flow chart showing scheduled enrolment, interventions and assessments of participants can be seen in figure 3.

## Patient involvement

Patient and members of the public were involved at several stages of the trial, including the design, management and conduct of the trial. We received input from stroke survivors in the design of the trial materials and management oversight through membership of the trial steering committee (TSC). We carefully assessed the burden of the trial interventions on patients. We continue to have patient involvement with the trial through representation on the steering committee and the investigators team. We will seek wider patient and public involvement in the interpretation of the trial findings and in development of an appropriate method of dissemination.

## Statistical methods and analysis
### Sample size
With 23 clusters per arm and an average of 20 patients per cluster, assuming an intraclass correlation of 0.03, a typical coefficient of variation of the cluster size of 0.65[22] and 2.5% significance (adjusted to 2.5% because of the use of two coprimary outcomes), we would be able to detect an effect size of 0.33 with at least 90% power on the coprimary outcomes (emotion and handicap subscales of the SIS v3.0[15]). The sample size calculation has been inflated to allow for a rate of 20% loss to follow-up for patients within clusters. Loss to follow-up of entire clusters is not anticipated.

### Analysis of primary outcome
We will use intention to treat methods for the analysis of the primary endpoints. A mixed effects model will be used to model each of the coprimary outcomes with a cluster random effect and fixed effects for the intervention and covariates that might potentially confound the relationship. Distributional assumptions will be assessed graphically by residual q-q plots and residual by fitted value plots. To handle the coprimary outcomes, 97.5% CIs will be reported for the two primary treatment effects which are equivalent to having the Bonferonni correction on the planned 5% significance level for a single endpoint.

Missing data will be analysed under the assumptions of missing completely at random and missing at random. Multiple imputation will be used to impute missing outcome data and the various potential predictors of missingness will be included in the imputation model.

Secondary analysis will look at the effect of time since stroke on uptake and effectiveness of the intervention.

### Economic evaluation
The cost-effectiveness (cost-utility) of the new system of care (intervention package) compared with usual care will be determined in a within-trial economic evaluation. Data will be collected via electronic primary care records and patient questionnaires on resource use implications of the intervention (including training), primary care visits, secondary care inpatient and outpatient visits, investigations, medications and use of social services. Patient and carer-incurred costs will also be considered to allow analysis from a broader societal perspective. Data collection will be undertaken within the trial to determine the time taken to deliver the structured review, and any additional resources required. Attendance at the individual and group MLAS sessions will also be recorded for every participant, and each session will be costed, taking into account staff time, any consumables and use of the venue. Standard unit costs will be applied to healthcare resource use including National Health Service (NHS) reference costs, the British National Formulary for medications and Unit Costs for Health and Social Care (Personal Social Services Research Unit).

The main outcomes of interest from the trial are quality of life (measured using EQ-5D-5L[18] at baseline and 6 and

12 months after entry into the trial) and capability (using the ICECAP-A questionnaire).[19] Initially, a cost-consequence analysis will be performed, to present a disaggregated analysis of all mean resource use and costs related to the intervention and usual care, healthcare, social care, patient/carer costs and EQ-5D-5L and ICECAP-A scores at all time points. Quality-adjusted life years (QALY) will be calculated by the area under the curve method using responses at all time points, and adjusted for baseline covariates including EQ-5D-5L score. Multiple imputation will be undertaken where there are missing cost and outcome data. An incremental cost-utility analysis will then be undertaken to determine the cost per QALY gained of the intervention compared with usual care.

To explore uncertainties in the analyses, deterministic sensitivity analysis is proposed to test the robustness of the results when varying key assumptions (eg, length of time required to deliver the intervention).

### Process evaluation
A process evaluation will examine the implementation of the IPCAS trial using both quantitative and qualitative methods. As well as capturing process variables, the evaluation will also entail a multidimensional approach to assessing intervention fidelity—the extent to which an intervention is delivered as planned.[23] Using the US National Institutes of Health Behaviour Change Consortium (NIHBCC) guidance[24] we will conduct a 'whole picture' assessment of the intervention across five fidelity dimensions: (1) design, (2) training, (3) delivery, (4) receipt, and (5) enactment. An overview is provided below, with the full protocol reported elsewhere (currently in submission).

Fidelity of design will be assessed through mapping intervention components to its purported theoretical frameworks. All intervention components have been specified a priori and recorded. Additionally, treatment differentiation (ie, extent to which intervention and control group practices differ) is considered by comparing the contents of the intervention versus usual care. Fidelity of training will be assessed using self-complete questionnaires (MLAS), video-recorded observations (MLAS) and audio-recorded observations (IPCAS). Fidelity of delivery will be assessed through audio-recorded observations (IPCAS), structured telephone calls to healthcare professionals and direct observations (MLAS). In addition, semistructured interviews will be conducted with healthcare professionals delivering the intervention, which will help assess both training and delivery. Fidelity of receipt and enactment will be assessed using self-complete questionnaires (MLAS), structured telephone calls and semistructured interviews with participants.

### Analysis
Quantitative aspects of the process evaluation (eg, process variables, coded video-recorded observations, self-complete questionnaires) will be synthesised descriptively. This will include what factors predict intervention fidelity. Qualitative aspects of the process evaluation (eg, semistructured interviews, qualitative data from questionnaires) will be synthesised using deductive thematic analysis, using the specific domains from the NIHBCC guidance.

### Reporting adverse events
We are not anticipating any intervention-related adverse events. Nevertheless, in accordance with Good Clinical Practice (GCP), each principal investigator is responsible for reporting all non-exempt serious adverse events (SAE) to the chief investigator (CI) within 24 hours of first notification. The CI is responsible for ensuring the assessment of all SAEs for expectedness and relatedness is completed and the onward notification of all non-exempt SAEs to the sponsor within 24 hours of first notification.

### Trial management
The trial is cosponsored by NHS Cambridgeshire and Peterborough Clinical Commissioning Group and the University of Cambridge. The study team work with local Clinical Research Networks in the East of England and the East Midlands to identify and recruit general practices.

Oversight of the trial will fall to an independent committee fulfilling the combined roles of TSC and data monitoring committee (DMC). They will provide overall supervision of the conduct of the trial on behalf of the trial sponsor(s) in accordance with National Institute for Health Research recommendations.[25 26] There are no prespecified criteria for electively stopping the trial prematurely. In the event that the joint TSC/DMC raise concerns over the safety of participants or the scientific integrity of the trial, a decision as to whether to continue will be discussed and voted on in keeping with the Terms of Reference of the committees and with GCP in Research guidelines.

### Data management and storage
Data completed by participants, such as consent forms and questionnaires, will be returned to the study team via post using prepaid stamped addressed envelopes. All relevant data collected at practice sites will be sent to the study team by trained and delegated practice staff via a secure transfer server. Paper data will be stored in locked filing cabinets within a security card-protected building at the University of Cambridge. Electronic data (including audio recordings) will be stored on a Secure Data Hosting Service protected by a dual authentication located on a firewall-protected virtual network (virtual LAN). Access to study data is restricted to the study team by dual authentication and group permissions. All investigators and trial site staff involved in this trial will comply with the requirements of the General Data Protection Regulation (EU) 2016/679 with regard to the collection, storage, processing and disclosure of personal information.

## ETHICS AND DISSEMINATION

Favourable ethical opinion for the research was gained on 19 December 2017 from Yorkshire and the Humber-Bradford Leeds NHS Research Ethics Committee. Approval to start was given by the Health Research Authority on 21 December 2017, prior to the recruitment of participants at any NHS site.

Patient and public involvement helped develop the dissemination plan. Data will be presented at national and international conferences and published in peer-reviewed journals.

**Collaborators** On behalf of the IPCAS investigator team: Marian Carey, Melanie Davies, Yvonne Doherty, Kamlesh Khunti, Lisa Lim, Bundy Mackintosh, Adrian Mander, Christopher McKevitt, Martin Roland, Stephen Sutton, Marion Walker, Elizabeth Warburton.

**Contributors** The study was conceived and designed by JM and RM. RM and JM drafted the manuscript with contributions from MRJA (process evaluation, intervention fidelity), SND (statistical analysis), VJ (design/description of MLAS programme), SJ (health economic evaluation) and EK (training of clinical staff, participant recruitment). All authors contributed to the design of the trial and review of the final manuscript. The IPCAS investigator team supported the design of the trial and provided comments/revisions to the writing of the final manuscript.

**Funding** This study was funded by the National Institute for Health Research's Programme Grant for Applied Research titled 'Developing primary care services for stroke survivors' reference PTC-RP-PG-0213 20001. The chief investigator for the study is JM, University of Cambridge email: jm677@medschl.cam.ac.uk. The IPCAS trial is cosponsored by the University of Cambridge and NHS Cambridgeshire and Peterborough Clinical Commissioning Group. This research is covered by the Cambridge University's Public Liability and Professional Indemnity policy.

**Disclaimer** The views expressed are those of the authors and not necessarily those of the NHS, the NIHR or the Department of Health.

**Competing interests** None declared.

**Patient consent for publication** Not required.

**Ethics approval** Favourable ethical opinion for the research was gained on 19 December 2017 from Yorkshire and the Humber-Bradford Leeds NHS Research Ethics Committee. Approval to start was given by the Health Research Authority (HRA) on 21 December 2017, prior to the recruitment of participants commencing at any NHS site. Patient recruitment started in March 2018. IRAS project ID: 233891. Protocol number: RG71908. REC reference: 17/YH/0441.

**Provenance and peer review** Not commissioned; externally peer reviewed.

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
