## [Reviewer comments · BMJ Open]

ARTICLE DETAILS

TITLE (PROVISIONAL)	The IPCAS Trial (Improving Primary Care After Stroke): protocol of a randomised controlled trial to evaluate a novel model of care for stroke survivors living in the community.
AUTHORS	Mullis, Ricky; Aquino, Maria Raisa Jessica (Ryc); Dawson, Sarah; Johnson, Vicki; Jowett, Sue; Kreit, Elizabeth; Mant, Jonathan

VERSION 1 – REVIEW

REVIEWER	Jenni Murray Bradford Institute for Health Research
REVIEW RETURNED	08-Apr-2019

GENERAL COMMENTS	The protocol is clearly written, comprehensive and appears to meet all the requirements of the Journal. I have a few minor comments that the authors may wish to address. I understand that a small pilot study has already been conducted but this isn't mentioned in the introduction, so we have no real sense of their insight into the limitations of the study, other than standard methodological ones. It looks like the intervention is targeted at stroke patients regardless of duration since the event. Many trials tend to dichotomise the targeting of their interventions to those early or later post-stroke (or even first ever stroke), which usually has a clear rationale given the trajectory of recovery and experiences after stroke. I'm sure the team discussed this and made a pragmatic decision to include all stroke patients regardless of duration since stroke; this makes sense in general practice. However, uptake of a self-management course may be very low in those who are beyond the first year of stroke for whom a settled routine is in place. Fidelity to the intervention therefore may be low in this group. It may be that the pilot didn't show this but we don't know. Could the authors tend to this point? Could they make it clear that their process evaluation will capture this information? Understanding who to target and how, in this intervention, will be valuable information for the already overburdened primary care team. Could the authors justify why they needed to develop their own self-management programme? Perhaps they could quote relevant systematic reviews and meta-reviews to support their rationale. Please provide a bit more detail in Figure 1. (Flow diagram) regarding timelines.
--

	The patient and public involvement section reads as if the study has already been completed. Please amend. The process evaluation will be very interesting given that there are, in effect, two interventions at play, targeting both staff and patients. The authors mention that they will map the intervention components to their theoretical frameworks. Could they provide a logic model for the intervention? Understanding what each component brings to the package in terms of proposed mechanisms and outcomes would be helpful. The references are missing a few dates.
--	--

REVIEWER	Alessio Baricich MD, PhD Assistant Professor Physical and Rehabilitation Medicine Università del Piemonte Orientale, Novara, Italy
REVIEW RETURNED	20-Apr-2019

GENERAL COMMENTS	In my opinion, this study protocol is scientifically credible, focusing on a crucial challenge for the Primary care and community services. The aim of the study and the Methods sections are adequately reported. However, it should be noted that the Authors refer to a "usual care", which is not clearly described. I understand that this study will involve a huge number of patients, in a large geographical area. However, considering the background of this study (e.g. the lack of a long-term program for these persons), the Authors should consider to briefly summarize this "usual care" model in order to better understand the differences compared to the IPCAS protocol.
---

REVIEWER	Associate Professor Dr Aznida Firzah Abdul Aziz Universiti Kebangsaan Malaysia (National University of Malaysia), Kuala Lumpur, Malaysia
REVIEW RETURNED	23-Apr-2019

GENERAL COMMENTS	The study is an important assessment of longer-term stroke care delivery at community level, albeit it is not as novel idea as claimed by the investigators. Please refer to the comments below. Overall, the proposal is fairly well written. Some further clarification is required as some points are overlooked i.e. this being a trial-within-trial study. Pg 3 lines 33-36 This statement is incorrect. There have been primary care-based model of care which have been described, and published. See ICARUS and iCaPPS. In fact, the protocol of this study closely resembles iCaPPS. Joubert, J., Davis, S. M., Donnan, G. A., Levi, C., Gonzales, G., Joubert, L., & Hankey, G. J. (2019). ICARUSS : An effective model
--

	for risk factor management in stroke survivors. Int J Stroke, 0(0), 1–16. https://doi.org/10.1177/1747493019830582 Abdul Aziz, A. F., Mohd Nordin, N. A., Ali, M. F., Abd Aziz, N. A., Sulong, S., & Aljunid, S. M. (2017). The integrated care pathway for post stroke patients (iCaPPS): a shared care approach between stakeholders in areas with limited access to specialist stroke care services. BMC Health Services Research, 17(1), 35. https://doi.org/10.1186/s12913-016-1963-8 Pg 4 lines 40-50: Suggest to separate the inclusion criteria for the GP practice and the patients. Will you be including patients with TIA or patients with Traumatic brain injury as well? How do you confirm the diagnosis of stroke? How will the sampling at GP surgery be conducted? Pg 7 lines 4-5: What are the components of 'usual care" for stroke patients at GP practices. It is important that this is listed as 'usual care" for patients in UK is not the same as in other parts of the world, and would make comparison of your study findings to be misinterpreted, particularly when looking at economic evaluation of stroke care services. Pg 7 lines 53-55: What assumptions are used to justify NHS reference costs for standard unit costs? Are all GP surgeries homogenous in terms of staffing, utilities and services provided? Pg 8 lines 47-49: What SAEs are expected / anticipated from this trial?
--	---

VERSION 1 – AUTHOR RESPONSE

Reviewer(s)' Comments to Author:

Reviewer: 1

I understand that a small pilot study has already been conducted but this isn't mentioned in the introduction, so we have no real sense of their insight into the limitations of the study, other than standard methodological ones.

We have added a sentence to this section to reflect this aspect of the intervention development: "The components of the model have been assessed for feasibility of delivery within primary care across four general practices prior to starting the IPCAS trial. This led to several minor procedural amendments aimed at improving implementation of the intervention."

It looks like the intervention is targeted at stroke patients regardless of duration since the event. Many trials tend to dichotomise the targeting of their interventions to those early or later post-stroke (or even first ever stroke), which usually has a clear rationale given the trajectory of recovery and experiences after stroke. I'm sure the team discussed this and made a pragmatic decision to include all stroke patients regardless of duration since stroke; this makes sense in general practice.

We have added a sentence to Setting and Participants and to the Analysis section to clarify our approach to this point:

"Given that primary care addresses the long term needs of stroke survivors we did not restrict participants to any specific time interval after their stroke".

“Secondary analysis will look at the effect of time since stroke on uptake and effectiveness of the intervention”.

However, uptake of a self-management course may be very low in those who are beyond the first year of stroke for whom a settled routine is in place. Fidelity to the intervention therefore may be low in this group. It may be that the pilot didn't show this but we don't know. Could the authors tend to this point? Could they make it clear that their process evaluation will capture this information?

Understanding who to target and how, in this intervention, will be valuable information for the already overburdened primary care team.

We agree this to be a key aspect in any potential roll-out of the intervention if proven effective. This data will be captured in our process evaluation and assessment of intervention fidelity, and will be described in detail in a separate publication. We have added a sentence to our brief description of the process evaluation to clarify this:

“This will include what factors predict intervention fidelity.”

Could the authors justify why they needed to develop their own self-management programme?

Perhaps they could quote relevant systematic reviews and meta-reviews to support their rationale.

We now cite three relevant systematic reviews (10, 11, 12) and have added the following sentence to the background:

“Systematic reviews have demonstrated that self-management after stroke shows promise, but evidence on aspects such as mood and social tasks remain sparse, with wide confidence intervals around effects on outcomes such as quality of life”.

Please provide a bit more detail in Figure 1. (Flow diagram) regarding timelines.

We have added more detail to the timeline in figure 1.

The patient and public involvement section reads as if the study has already been completed. Please amend.

We have added the following to the PPI section of the text:

“We continue to have patient involvement with the trial through representation on the Steering Committee and the investigators team. We will seek wider patient and public involvement in the interpretation of the trial findings and in development of an appropriate method of dissemination.”

The process evaluation will be very interesting given that there are, in effect, two interventions at play, targeting both staff and patients. The authors mention that they will map the intervention components to their theoretical frameworks. Could they provide a logic model for the intervention? Understanding what each component brings to the package in terms of proposed mechanisms and outcomes would be helpful.

We have now included a logic model for the IPCAS trial intervention (figure 2).

The references are missing a few dates.

These have been added or moved as required.

Reviewer: 2

Please leave your comments for the authors below

In my opinion, this study protocol is scientifically credible, focusing on a crucial challenge for the Primary care and community services.

The aim of the study and the Methods sections are adequately reported.

However, it should be noted that the Authors refer to a "usual care", which is not clearly described. I understand that this study will involve a huge number of patients, in a large geographical area.

However, considering the background of this study (e.g. the lack of a long-term program for these persons), the Authors should consider to briefly summarize this "usual care" model in order to better understand the differences compared to the IPCAS protocol.

We thank the reviewer for highlighting a lack of detail provided on usual care. We have added a sentence in the description to show how we intend to capture the key elements of stroke care provided by Practices randomised to the usual care arm of the trial:

"Currently no standard package of long-term care for stroke survivors exists in Primary Care, and therefore we expect "usual care" to vary between Practices. We will capture information on the key elements of care provided by each participating Practice to enable comparison between the two arms of the trial".

Reviewer: 3

The study is an important assessment of longer-term stroke care delivery at community level, albeit it is not as novel idea as claimed by the investigators.

Please see our response below regarding the nature of the IPCAS intervention.

Please refer to the comments below.

Overall, the proposal is fairly well written. Some further clarification is required as some points are overlooked i.e. this being a trial-within-trial study.

The IPCAS model of care comprises six elements, all of which are outlined in the protocol. The IPCAS trial is a two-arm cluster randomised design comparing the IPCAS model of care with usual current practice. We do not consider this to be a "trial-within-trial study". However, we will be undertaking a comprehensive evaluation of intervention fidelity to elucidate the contribution that each component makes.

Pg 3 lines 33-36

This statement is incorrect. There have been primary care-based model of care which have been described, and published. See ICARUS and iCaPPS. In fact, the protocol of this study closely resembles iCaPPS.

Joubert, J., Davis, S. M., Donnan, G. A., Levi, C., Gonzales, G., Joubert, L., & Hankey, G. J. (2019). ICARUSS : An effective model for risk factor management in stroke survivors. *Int J Stroke*, 0(0), 1–16. <https://doi.org/10.1177/1747493019830582>

Abdul Aziz, A. F., Mohd Nordin, N. A., Ali, M. F., Abd Aziz, N. A., Sulong, S., & Aljunid, S. M. (2017). The integrated care pathway for post stroke patients (iCaPPS): a shared care approach between stakeholders in areas with limited access to specialist stroke care services. *BMC Health Services Research*, 17(1), 35. <https://doi.org/10.1186/s12913-016-1963-8>

We are not aware of published trials that have evaluated Primary Care models of holistic long-term stroke care such as IPCAS. The ICURUSS trial focussed on vascular risk factor management early after stroke, and is not intended to address the wider health and well-being issues that stroke survivors have reported. As far as we are aware, the iCaPPS model is yet to be evaluated in clinical trials to assess its effectiveness when compared with current practices. However, we acknowledge that other work in this area is on-going and have amended the sentence in the abstract (Pg 3 lines 33-

36) to clarify that we are referring to holistic models of stroke care supported by evidence from clinical trials:

“No formal primary care based holistic model of care with clinical trial evidence exists to support stroke survivors living in the community, and stroke survivors report that many of their needs are not being met”.

We have also added this sentence to the background.

Pg 4 lines 40-50: Suggest to separate the inclusion criteria for the GP practice and the patients.

Thank you for this suggestion. We have described the inclusion criteria for the GP practice in the text, and the inclusion criteria for the patients as bullet points to enable the reader to distinguish between the two.

Will you be including patients with TIA or patients with Traumatic brain injury as well? How do you confirm the diagnosis of stroke?

General Practices in the UK have a stroke register of patients defined by specific diagnostic codes in their medical records. Quality of care for patients on the stroke register forms part of the NHS remuneration package, therefore every Practice has a pre-defined stroke register of patients. Patients with a diagnostic code of TIA only will appear on the stroke register, but are not included in the IPCAS trial as (by definition) they will have no residual morbidity from their vascular event. Although patients with traumatic brain injury may benefit from the IPCAS model of care, we have restricted inclusion in the trial to stroke to reduce heterogeneity within the cohort.

How will the sampling at GP surgery be conducted?

We have added the following text to the methods:

“If practices had 110 or fewer such people, invitations were sent to all those eligible. For larger practices, a random sample of 110 eligible patients were sent invitations”.

Pg 7 lines 4-5: What are the components of 'usual care" for stroke patients at GP practices. It is important that this is listed as 'usual care" for patients in UK is not the same as in other parts of the world, and would make comparison of your study findings to be misinterpreted, particularly when looking at economic evaluation of stroke care services.

Please see our response to reviewer 2 above.

Pg 7 lines 53-55: What assumptions are used to justify NHS reference costs for standard unit costs? Are all GP surgeries homogenous in terms of staffing, utilities and services provided?

The use of standard unit costs (e.g. NHS Reference Costs) is well-accepted and common place in UK-based health economic analyses, and their use has the advantage of having been developed through a common methodology. The values represent price weights, and whilst we are aware that there will always be variations between providers (e.g. GP surgeries), unit costs allow for consistency across studies which address UK resource allocation decisions.

Pg 8 lines 47-49: What SAEs are expected / anticipated from this trial?

We are not anticipating any adverse events relating directly to the IPCAS intervention. We have added the following to our paragraph on these in the Reporting Adverse Events section:

“We are not anticipating any intervention-related adverse events. Nevertheless, in accordance with Good Clinical Practice (GCP)...”

VERSION 2 – REVIEW

REVIEWER	Dr Jenni Murray Bradford Institute for Health Research
REVIEW RETURNED	23-May-2019

GENERAL COMMENTS	I am satisfied with how the authors have responded to the comments. I have no further comments.
---

REVIEWER	Alessio Baricich Physical and Rehabilitative Medicine Department of Health Sciences. Università del Piemonte Orientale, Novara, Italy
REVIEW RETURNED	06-Jun-2019

GENERAL COMMENTS	This study protocol is scientifically credible, focusing on a crucial challenge for the Primary care and community services. In my opinion, it is now suitable for publication.
---

REVIEWER	Aznida Firzah Abdul Aziz Dept Family Medicine, Faculty of Medicine, Universiti Kebangsaan Malaysia, Kuala Lumpur, Malaysia
REVIEW RETURNED	07-Jun-2019

GENERAL COMMENTS	This is my second review of this proposal. the authors have not addressed any of my previous comments which I have raised in my earlier review. Hence, I feel the authors have no plans to take heed of any comments from me. As reiterated in my earlier, the study is NOT a novel terms of primary-care based long term stroke care provision at community level. Studies have been conducted in Australia and Malaysia. In fact the study protocol closely resembles the one in Malaysia- of which some have already been published. I have no further comments- and my final decision is to reject this proposal.
--